# The Breadth of Bacteriophages Contributing to the Development of the Phage-Based Vaccines for COVID-19: An Ideal Platform to Design the Multiplex Vaccine

**DOI:** 10.3390/ijms24021536

**Published:** 2023-01-12

**Authors:** Ihtisham Ul Haq, Katarzyna Krukiewicz, Galal Yahya, Mehboob Ul Haq, Sajida Maryam, Rasha A. Mosbah, Sameh Saber, Mohammed Alrouji

**Affiliations:** 1Department of Biosciences, COMSATS University Islamabad (CUI), Islamabad 44000, Pakistan; 2Department of Physical Chemistry and Technology of Polymers, Silesian University of Technology, M. Strzody 9, 44-100 Gliwice, Poland; 3Joint Doctoral School, Silesian University of Technology, Akademicka 2A, 44-100 Gliwice, Poland; 4Centre for Organic and Nanohybrid Electronics, Silesian University of Technology, Konarskiego 22B, 44-100 Gliwice, Poland; 5Department of Microbiology and Immunology, Faculty of Pharmacy, Zagazig University, Al Sharqia 44519, Egypt; 6Infection Control Unit, Zagazig University Hospital, Zagazig University, El Sharkia 44519, Egypt; 7Department of Pharmacology, Faculty of Pharmacy, Delta University for Science and Technology, Gamasa 11152, Egypt; 8Department of Clinical Laboratory Sciences, College of Applied Medical Sciences, Shaqra University, Shaqra 11961, Saudi Arabia

**Keywords:** phages, COVID-19, therapeutics, variants, phage display technology, vaccines, CRISPR

## Abstract

Phages are highly ubiquitous biological agents, which means they are ideal tools for molecular biology and recombinant DNA technology. The development of a phage display technology was a turning point in the design of phage-based vaccines. Phages are now recognized as universal adjuvant-free nanovaccine platforms. Phages are well-suited for vaccine design owing to their high stability in harsh conditions and simple and inexpensive large-scale production. The aim of this review is to summarize the overall breadth of the antiviral therapeutic perspective of phages contributing to the development of phage-based vaccines for COVID-19. We show that phage vaccines induce a strong and specific humoral response by targeted phage particles carrying the epitopes of SARS-CoV-2. Further, the engineering of the T4 bacteriophage by CRISPR (clustered regularly interspaced short palindromic repeats) presents phage vaccines as a valuable platform with potential capabilities of genetic plasticity, intrinsic immunogenicity, and stability.

## 1. Introduction

The scientific focus has been concerned with effective vaccine development to quell the ongoing COVID-19 pandemic. Vaccine development is usually considered a long investment from discovery to licensure [1]. COVID-19 emerged in China [2] and was later declared a pandemic in March 2020 [3]. According to the World Health Organization (WHO), more than 550 million cases were confirmed globally, with more than 6.3 million deaths reported so far. There were tremendous efforts for the early detection of COVID-19, the identification of severity biomarkers [4], and the design of effective therapeutics against COVID-19 employing different approaches, such as the in silico screening of previously approved therapeutics and phytochemicals or the rational design of novel drugs against the crucial enzymes of the SARS-CoV-2 virus [5,6,7,8,9,10]. Effective vaccines against COVID-19 were designed and achieved remarkable feedback on COVID-19 transmission, hospitalization, and mortality [11]. Despite these alarming situations, the world has not reached 70% vaccination coverage for the global human population [12]. Thereby, different strategies for vaccine development were employed worldwide to overcome this highly contagious and rapidly evolving virus posing challenges to attaining the “One Health” goal [13].

Phage research implies a dynamically emergent platform whose scope is constantly expanding in the biomedical sciences. The therapeutic potential of phages was discovered by Felix d’Herelle a century ago and continued to serve as an ideal platform for the development of a broad array of molecular tools that contributed to cracking some important mysteries in biology [14]. Over the past few years, phage research delivered incredible breakthroughs in the field of biomedical sciences, which caused a remarkably intense upturn in its exploitation in therapeutics, particularly for vaccine development [15]. Phages represent a powerful platform, having a substantial extent in producing therapeutically important antibodies with desired binding characteristics and serving as immunotherapies against several infectious diseases [16]. Phages were explored as antiviral vaccine delivery vehicles in life-threatening infectious diseases [17,18]. The phage-based vaccine provides sufficient protection against several viral diseases, such as human immunodeficiency virus (HIV) [19], herpes simplex virus 1 (HSV-1) [20], and human papillomavirus type 16 (HPV16) [21,22]. Moreover, phage-based vaccines were also reported to trigger robust immune responses against a variety of displayed antigens of the influenza virus [23] and the Middle East respiratory syndrome coronavirus (MERS-CoV) epidemics [24].

The emergence and circulation of multiple variants of SARS-CoV-2 and the global rollout of new vaccines compel the need for universal and multiplex vaccine design platforms that are pliable to rapid tuning and steady preparation to simplify the engineering of vaccine delivery [25]. Various vaccine candidates were considered for vaccine development during the COVID-19 pandemic [26]. The protection efficiency and vaccine efficacy diverging towards the new emerging variants of SARS-CoV-2 [27] is a serious concern in COVID-19 treatment. The available mRNA SARS-CoV-2 vaccines are efficacious in preventing COVID-19 disease; however, they do not provide sufficient protection and are unable to persuade proficient mucosal immunity [28]. Effective vaccines could be discovered through universal and multiplex vaccine design platforms, which should accede to the inclusion of multiple targets, multi-component and full-length proteins, peptides, and DNA vaccines. Consequently, the vaccine discovery timeline could be compressed by such a universal platform that negotiates the hazardous selection of the most effective vaccines lacking iterative design cycles [29].

Such scenarios attract a vaccine development focus towards phages to design multiplex vaccines for COVID-19. This article summarizes the overall breadth of the antiviral therapeutic perspective of phages contributing to the development of phage-based vaccines for COVID-19. It should provide a concise and precise description of the experimental results, their interpretation, as well as the major conclusions that could be used for further research in the field of phage-based vaccines.

## 2. Phages: An Ideal Platform to Design Multiplex Vaccines

Phages are found in almost all sorts of biological habitats, thus being the highly ubiquitous biological agent that renders them ideal tools for molecular biology and recombinant DNA technology. This allows the possibility to genetically modify them to express peptides or proteins on their double-stranded or single-double-stranded genome and the shape of their capsid protein. Despite the great variation, 95% of them are non-enveloped, dsDNA, tailed phages [30]. In the majority of vaccine development cases, the capsid of dsDNA-tailed phages is used to express a viral antigen or a portion of an antigen, most typically via non-covalent linkages [31]. The concern of a significant reduction in the efficacy of existing COVID-19 vaccines due to rapid viral mutations could be resolved by the development of phage vaccines, such as phage displayed vaccines, phage DNA vaccines, and hybrid vaccines [32]. The subsequent phage vaccines created are presently accepted, assembled, practice-invented, and undergoing extra preclinical trials, such as the optimization of the dosage, administration route, and immunity extensiveness [28].

Phage DNA vaccines are prepared by the incorporation of a eukaryotic expression cassette with the gene encoding an antigen or by mimicking the epitope of the phage genome [33]. Phage displayed vaccines deliver peptides or proteins, particularly immunogenic ones, to the target cells or tissues. These immunogenic peptides, proteins, and chemically conjugated antigens are displayed on phage surfaces [34], as shown in Figure 1. Phage displayed vaccines and phage DNA vaccines are combined to prepare hybrid vaccines. Intensive work on the phage DNA vaccines against COVID-19 is proceeding [35]. Several antigens do not permit their correct folding in conventional DNA and phage displayed vaccines. In this scenario, phage DNA vaccines could be valuable alternatives that exclude the shortcomings of other vaccines. Various remunerations of the phage vaccines may come from the symmetric, tedious, and wide varieties of antigens presented on the capsid of phage that imitate the PAMPs (pathogen-associated molecular patterns) of pathogenic viruses [36]. Several features, such as high stability, low reactivity, safety, low cost for mass production, easy genetic manipulation, and the rapid identification of target proteins, render the phage an ideal candidate for the development of COVID-19 vaccines [36]. The phage-based delivery systems of antigens are shown in Figure 1.

Phages induce antiviral cytokines, such as IFN-α and IL-12, and trigger the TLR3-dependent pattern recognition receptors and the inhibition of TNF-driving type I IFN [37,38]. Additionally, phages have the potential to trigger innate and adaptive immune responses in humans without causing any adverse effects [39]. It was reported that antiviral immunity induced by phages conferred the suppression of the activation of the nuclear factor kappa-light-chain-enhancer of activated B cells [40]. Phages were also reported to induce antiviral immunity by upregulating defensin 2 [40], which is effective in preventing viral infections.

The studies [41,42] pointed out that the phage vaccine’s efficacy against SARS-CoV-2 recombinant antibodies could be produced efficiently with phage display techniques; previously, they were also proven successful in the Middle East respiratory syndrome coronavirus (MERS-CoV) epidemics [23]. All the available COVID-19 vaccines are efficacious in preventing the disease. However, they do not provide sufficient protection and are unable to persuade proficient mucosal immunity [43]. In such a scenario, phage vaccines could be proven as an ideal platform to cope with the COVID-19 pandemic. The induction of phage-conferred antiviral immunities suggested the use of phage-based vaccines for COVID-19 [44]. The ability of phages to provoke antibodies (humoral immunity) and T cell responses (cellular immunity) has imperative consequences for phage therapy and phage display vaccines [36]. Phage-conferred adaptive immunity is presented in Figure 2.

## 3. Phage Display Technology in Vaccine Development

The expression of any peptide or protein on the phage surface by fusing them to phage coat proteins is called phage display technology, which is now a powerful technology for vaccine development [46]. Recently, the phage display technique has increased in vaccine development as it facilitates a better understanding of the immunization process. It offers a platform that quests immunogenic peptide sequences and constructs new vaccines [47]. The introduction of phage display technology was a turning point in the history of phage research. Genomes that could be either DNA or RNA and the capsid are the two major components of the phage. Capsid genes are targets in phage display technology as desired genes may be inserted into capsid genes, and subsequently, after replication, the phage capsid would also display the characteristics of the inserted gene [24]. The peptides may be easily identified through phage display peptide library technology, which may specifically bind to an antigen [35]. The proteins or peptides translated from the gene of interest could be an integral part of any microorganism and may act as foreign antigens for humans [17].

## 4. Phage Display Technology in the Development of Antiviral Vaccines

Phages were explored as antiviral vaccine delivery vehicles in life-threatening infectious diseases [17]. The significant factor in antiviral vaccines is cytotoxic T-lymphocyte (CTL) [48,49]. Thus, in triggering CTL-specific responses, the abilities of the CTL phage displayed epitope were extensively studied. T cell antigen receptor (TCR) was isolated and enhanced from a person infected with the human immune deficiency virus (HIV). This TCR was specific for the SL9 peptide, an immunodominant HLA-A * 02-restricted, HIV Gag-specific SLYNTVATL sequence. Interestingly, all the escape variants of the epitope were recognized along with the targeting of HIV-infected cells using high-affinity TCRs [18]. It was reported that a phage DNA vaccine containing herpes simplex virus 1 (HSV-1) glycoprotein D cassette triggered humoral and cellular immune responses in BALB/c mice, thus showing the efficiency of phage vaccines against HSV-1 [19]. Phage-based antiviral vaccines were also effective against human papillomavirus type 16 (HPV16) and offered comprehensive protection against HPV infection. The VLP robust anti-HPV16 L2 serum antibodies were generated with the minor capsid protein L2 epitope displayed on phages [20,21,50]. Table 1 indicates the phage-based vaccine/immunizations with antiviral efficacies.

Bazan and colleagues [46] stated that phage display technology would be further used in the vaccine development against negative-stranded RNA viruses due to the emergence and high infectivity of these viruses. In the research of antiviral vaccines, five promising peptides were identified with broad-spectrum activities [51]. Moreover, phage display technology provides sufficient antiviral protection against human hepatitis C virus (HCV) and human hepatitis B virus (HBV) infection [38]. The MHC class I restricted hepatitis B-specific CTL response was induced in BALB/c (H-2d) mice inoculated with a hybrid phage containing the pVIII gene with the CTL epitope [52].

**Table 1 ijms-24-01536-t001:** Overview of phage-based vaccine/immunizations with antiviral efficacies against deadly pathogenic viral infections.

Phage Genera	Efficacy of Viral Phage-Based Vaccines	Reference
Filamentous phage particles	Phage DNA vaccine containing the herpes simplex virus 1 (HSV-1) glycoprotein D cassette triggered humoral and cellular immune responses in BALB/c mice, thus showing the efficiency of phage vaccines against HSV-1.	[20]
Bacteriophage PP7	Phage-based antiviral vaccines were also found effective against human papillomavirus type 16 (HPV16) and offered comprehensive protection against HPV infection.	[21]
T4 phage	Phage display technology provides sufficient antiviral protection against human hepatitis virus (HCV) and human hepatitis B virus (HBV) infection.	[46]
Hybrid phage	The MHC class I restricted hepatitis B-specific CTL response was induced in BALB/c (H-2d) mice inoculated with a hybrid phage containing the pVIII gene with the CTL epitope [44].	[52]

## 5. The Consideration of Phage Display Technology in COVID-19

Rao’s lab explored the biomedical applications of phage display technology and also successfully designed dual anthrax-plague vaccines with phage display technology [28]. They also used the gene-editing tool ‘‘CRISPR’’ to rapidly construct recombinant phages for the development of a COVID-19 vaccine [53]. After the declaration of the COVID-19 pandemic, the majority of labs closed with the worldwide lockdown; however, Rao’s lab shifted their work to therapeutic medications of SARS-CoV-2 instead of shutting down. They successfully developed phage-based vaccines, which were capable of inhibiting SARS-CoV-2 infections both in vitro and in vivo. Mice were immunized with these vaccines, which showed no symptoms of SARS-CoV-2 infection, and the vaccines were also capable of blocking SARS-CoV-2 infection with in vitro experiments [18].

## 6. Bacteriophages: A Tool for the Expression and Presentation of SARS-CoV-2 Antigens

SARS-CoV-2 is a rapidly evolving virus [54,55,56,57,58], and in the last two years, many variants of the virus have been observed; thus, vaccination alone is insufficient to overcome this pandemic, and thereby cost-effective and reliable methods to understand the mechanisms of viral pathogenicity and immunogenicity in the host are crucial. In a recent study, the SARS-CoV-2 spike S1, the protein’s phage mimotopes which resemble the shape of the viral receptor-binding (RB) sites, were created. Utilizing recombinant bacteriophages with surface mimotopes may provide a molecular probe for analyzing viral infectivity and could be effective in COVID-19 vaccine development [59]. Another study used mycobacteriophages to show SARS-CoV-2 RB motif segments. Numerous recombinants exhibiting peptides at high densities on the phage capsid were created and demonstrated that they might elicit IgG-mediated antibody responses in mice against the SARS-CoV-2 epitopes [23]. Some reports have highlighted the binding affinity of the 12-mer phage display peptide against the RB domain of the S protein [60]. Staquicini and colleagues [61] designed and developed a phage vaccine for COVID-19 by using dual ligand peptide-targeted phages and adeno-associated virus/phage (AAVP) particles which showed strong targeted humoral responses against the S protein and were considered promising candidates for COVID-19 vaccination [61] A strong systemic and specific immune response was observed with phage particles against SARS-CoV-2 in immune-competent mice by aerosol pulmonary vaccination [61]. The S protein gene of SARS-CoV-2 was inserted in the phage and displayed short epitopes on the phage capsid, showing the feat of phage vaccines [61]. A strong and specific humoral response is induced by these targeted phage particles carrying the epitopes of the SARS-CoV-2 spike (S) protein. Table 2 lists all the phage-based vaccines for COVID-19.

*E. coli* infecting phages, also known as T4 phages, belong to the Myoviridae family and serve as an extraordinary model organism in biotechnology and molecular biology and could certainly be used to prepare vaccines against the SARS-CoV-2 pandemic in a relatively short time. The T4 phage is considered one of the most effective vaccine delivery platforms; however, previously, it was deliberated to lack the flexibility of engineering for the rapid generation of multiplex vaccine candidates against the SARS-CoV-2 pandemic [62]. Another study described the design and development of a monovalent lambda phage-like particle-based vaccine against coronaviruses. A robust humoral immune response was observed in mice models after the administration of the vaccine, and a very effective immune response was observed [28]. Likewise, Zhu and colleagues developed a T4-based mucosal vaccine prototype by designing spike trimers on the exterior surface of the protein and the interior surface of the nucleocapsid protein. The vaccine showed valid results in providing complete protection against COVID-19 [40]. Different parts of the T4 phage genome have an enormous amount of unnecessary genetic space that could be ideally used for the development of a universal vaccine design template. Phage-based vaccines induce humoral and cell-mediated host immune responses, both chemically stable and economical, compared to conventional vaccines [15]. The immune system stimulating potential is shown in Figure 3.

Recently, the efficacy and protection of the ability of monovalent and bivalent phage-like particle vaccines were determined using the receptors binding domain of the MERS and SARS-2 coronaviruses spike proteins [28]. The study had both in vitro and in vivo features to evaluate the immunoreactivity and immunogenicity potential of the phage-like particle vaccines derived from the lambda bacteriophage. Two doses of vaccines were administered to a female mouse, one on the first day and the second after three weeks, and subsequently, blood was collected for the immunological assays. A promising antibody induction with strong neutralizing activity was observed and consequently provided durable protection against both viruses. The mouse intentionally infected with SARS-CoV-2 did not develop any lung disease symptoms after immunization with the phage-like particle vaccines. Thus, presenting a “designer nanoparticle” platform with the flexibility and sturdiness of phage-like particle coronaviruses vaccines that offered promising therapeutic interventions [28]. We conclude that multitarget COVID-19 vaccines could be produced rapidly through a phage-like particle platform.

## 7. CRISPR Engineering in Phage Vaccines against COVID-19

Very recently, viral components of SARS-CoV-2, such as envelope and capsid proteins, were inserted into a phage using CRISPR engineering, as shown in Figure 3. Segments of the SARS-CoV-2 genome were inserted in *E. coli* strains using the T4 genome phage, and each strain consisted of dual plasmids. Type 2 Cas9 and 5 Cas12a showed the genome-editing nucleases of the spacer plasmid. Furthermore, the protospacer sequences in the genome phage have a second plasmid (donor) which also consists of the SARS-CoV-2 sequence that is the target for the CRISPR RNA (crRNAs or the space RNA) [63]. The latter has homologous arms in the flanking position of approximately 500 base pairs long, used for the point of insertion. The genome has four non-essential regions that are used for the insertion of several SARS-CoV-2 genes. A universal vaccine platform that presents various important features was developed using phage T4 nanoparticles and CRISPR engineering with the exceptional advantage of generating rapid vaccines in any viral pandemic, including COVID-19 [28]. The use of type II Cas9 and type V Cas12a nucleases with CRISPR genome engineering created a sequence of recombinant SARS-CoV-2 gene insertions in a phage for the development of the T4 COVID-19 vaccine. In trials, antibody responses were broadly provoked by the T4 COVID-19 vaccine against various components, such as NP-specific and E-specific antigens, which confirmed the efficacy of the T4 COVID-19 vaccine [63]. During natural SARS-CoV-2 infection, T cells of the host immune system target NP-specific antigens [28]. Thus, specific and broader immune responses are triggered against multiple viral targets using a single-phage backbone in which various antigens are incorporated through the T4 CRISPR platform, as shown in Figure 4.

## 8. The Pharmacological Attributes of Phage Vaccines in COVID-19

The potential advantage of phage-based vaccines is the establishment of a staged approach with the capability of rapid modification in the vaccine in response to mutations. The induction of a broader immune response by a single-phage backbone renders it one of the best vaccine candidates to reduce the vaccine escape mutants significantly [62]. There is no need for adjuvants in phage vaccines, and the latitudinal nature of the exposed epitopes could be manipulated easily by the genetic and structural engineering of phages. The desired antigen could be combined with the phage backbone to create different formulations of the phage vaccine [63]. The diversity of the immune response could be increased in the same formulation carrying distinct and multiple antigens [64]. Phage display is a quick way to identify antibodies directed against any antigen of interest, and, as a result, this technology is already being used for the development of therapeutic antibodies. This necessitates the use of DNA from the beta cells isolated from people who have already developed the necessary antibodies. The European Union and the United States of America are both running initiatives to collect convalescent plasma with antibodies that target SARS-CoV-2 [65].

It is estimated that a few hundred thousand to 1,000,000 phage vaccine doses against SARS-CoV-2 could be formed in a 100-L fermenter, thus leading to large-scale production. Phage vaccines would be a solid possibility for extensively defensive COVID-19 vaccines and booster immunizations to previously vaccinated people [28,66]. Conventional vaccines developed against COVID-19 are generally effective in the prevention of infectious diseases, but their lack of stability raises concerns with transport, safety, and targeted delivery [67]. Phages are exceptionally stable nanoparticles, have a great profile of safety, and may be produced at moderately minimal costs. Phages are non-pathogenic to humans, have negligible reports of infections, and stimulate the eukaryotic immune system with potential adjuvant abilities [63]. They provide a new and strong elective stage to quickly produce viable antibodies against any pestilence or pandemic later on, especially when multivalent immunizations are crucial for controlling future pandemics and securing worldwide networks. The adaptability and flexibility of the phage vaccine are the potential benefits that could conquer the emerging variants of SARS-CoV-2. Phage vaccine efficacy could be increased even more by combining the spike and nucleocapsid proteins, and it would be more effective against the current and future variants of SARS-CoV-2 [68]. In the context of vaccine design, phage display is a fast, low-cost, reliable, and high-throughput method for the screening and selection of peptide antigens in an effective and straightforward manner [69]. These approaches are fast, inexpensive, highly adaptable, and easy to use.

## 9. Safety and Biocompatibility Concerns with Phage Vaccines

Favorable safety measures in phage vaccines are the key feature principally contributing to the therapeutic activities of phages [51], and the immunized organism is at lower risk of phage-conferred pathogenicity. The flexibility in the route of administration of the phage vaccine is another possible advantage, which could be either mucosal, intramuscular, or oral based on the adaptability to virus mutations [69]. Additionally, phages protect host cells by competing with the SARS-CoV-2 virus for surface assimilation and penetration, thus acting as a shield for eukaryotic cells. The pandemic burden could be reduced by phage-based vaccines whose safety profiles are well-recognized and not expensive compared to other vaccines [70]. Therefore, the burden of the emergence and circulation of SARS-CoV-2 variants could be reduced through the development of phage-based vaccines. Despite the successful presentation of peptides in phages, the major limitation of the development of these vaccines involves the accurate display of these molecules on the surface of the phage [51]. This problem revolves around two major aspects: the correct folding of a polypeptide chain and the sufficient display of functional epitopes. Choosing a peptide with the right strong antigenic characteristics is essential. However, a major difficulty is finding the right size of the peptide to be fused with the phage peptide without affecting the structure and other properties of the phage [71]. The efficiency of phage vaccines is dependent on both these properties in order to produce a significant immune response [47]. 

Despite being labeled as safe, there is an active possibility of bacterial infection in the gut if orally administrated. Phages may infect the bacteria in the gut, which could release harmful toxins to the host [72]. The size of the antigen should be small as large peptides cannot be accommodated into the tail tube or capsid; thus, a large antigen could be a hindrance in phage vaccine development. Moreover, the size of the genome should also be considered, as the length of the genome and the packaging capacity of the virion should overlap [73]. However, this limitation is evaded with certain other platforms [28]. Some studies have reported the alteration in the structure of the peptide upon insertion of the SARS-CoV spike epitopes into the phage. This poses another challenge in the production of phage-based vaccines. It was speculated that the epitopes with the minimum number of alterations in the normal structure are most efficient in producing the immune response [61]. In conclusion, the behavior of the immune system in response to such vaccines and the exhibition of peptides in the suitable confirmation for the maximum immunological response remain open questions for the successful designing of such vaccines.

Virus-like particles (VLPs)were also reported as an effective and safe platform for vaccine design and development. However, it is difficult to produce clinically viable VLPs [74]. Compared with subunit vaccines, VLPs have higher stability. Though, they are not stable during downstream processing and when environmental conditions change, as they lack viral genetic material [75]. There are stability concerns reported for VLP vaccines already available in the marketplace. The integrity of VLPs is altered due to the impact of temperature changes, agitation rates, fluid dynamics, and chemical treatments [76]. The immunogenicity of VLPs significantly reduces due to structural breakdowns. It also interferes with cell growth and the production of metabolic proteins, which impacts VLP production.

## 10. Conclusions and Future Recommendations

It is concluded that phages have considerable breadth in SARS-CoV-2 vaccines, which offer many significant advantages and represent phages as universal adjuvant-free nanovaccine candidates and single-phage scaffolds used for the inclusion of multiple targets. The genetic manipulation of phages allows the rapid production of multiplex vaccine candidates that could be effective against multiple variants of SARS-CoV-2 with broader immune responses. The engineering of the T4 bacteriophage by CRISPR (clustered regularly interspaced short palindromic repeats) presents phage vaccines as a substitute universal adjuvant-free nanovaccine platform with flexibility in route of administration and a high safety profile. Phages are exceptionally stable nanoparticles and have a great safety profile. The vaccine development, approval, and quality control of vaccine components take a long time and may go on for years or even decades to develop the final product. However, phage display technology represents a novel toolbox for vaccine innovation and shortlists the development time frame.

Future investigations need to optimize the phage vaccine platform to expand the diversity of phage vaccines for COVID-19 and prevent future pandemics. It provides a new and strong elective stage to rapidly produce viable antibodies against any pestilence or pandemic later on, especially when multivalent immunizations are crucial for controlling future pandemics and securing worldwide networks.

## Figures and Tables

**Figure 1 ijms-24-01536-f001:**
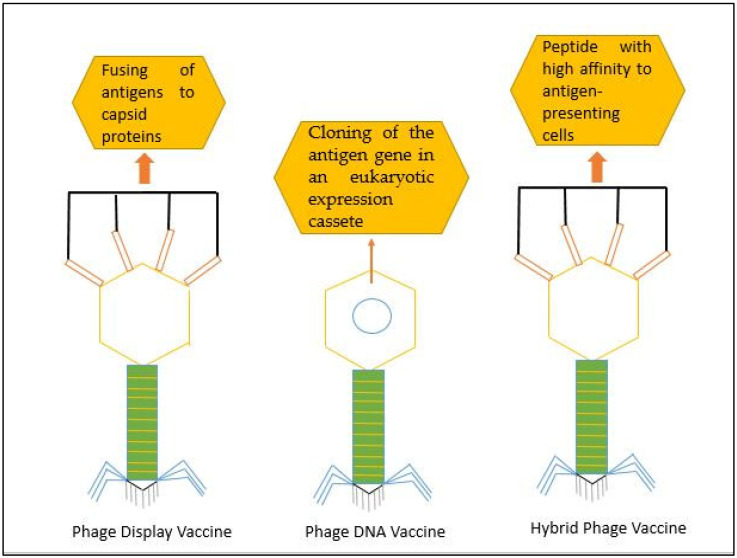
Basic representation of three different types of phage-based vaccines; phage display vaccines, phage DNA vaccines, and hybrid phage vaccines.

**Figure 2 ijms-24-01536-f002:**
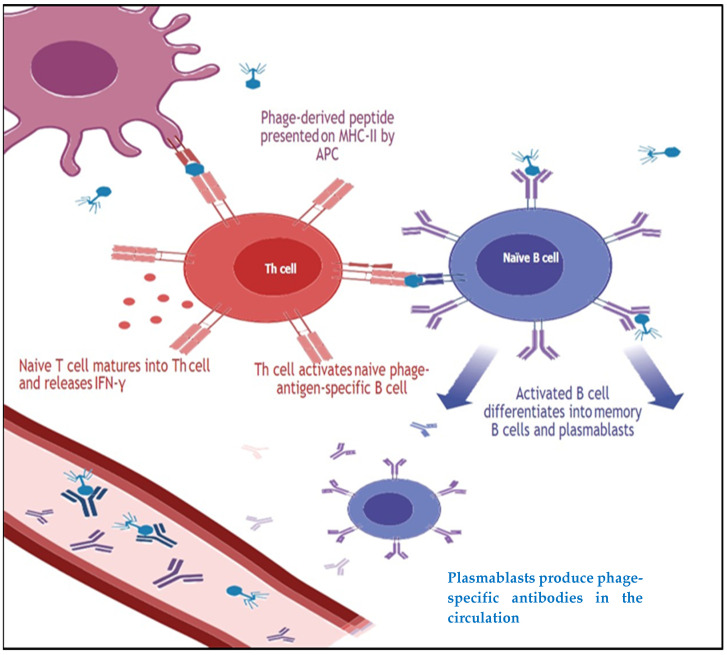
Bacteriophage interactions with adaptive immune cells. Phage-derived peptides are presented by APCs to naive T cells on MHC-II molecules. Naive CD4 + T cells specific for the antigen-MHC undergo activation (via the release of IFN-γ) and proliferation into Th cells. Phage-specific Th cells are then capable of activating naive phage-specific B cells. Activated B cells differentiate into plasmablasts, short-lived cells that circulate and produce large quantities of phage-specific antibodies, and memory B cells, which may be reactivated upon subsequent phage exposure and initiate the production of further antiphage antibodies. Antiphage antibodies bind and inactivate phages in the circulation and within tissues. Abbreviations: APC, antigen-presenting cell; IFN, interferon; MHC, major histocompatibility complex; Th, T helper [45]. Reproduced with permission from [Paul L Bollyky], [Annual Review of Virology]; published by [Annual Reviews], [2021].

**Figure 3 ijms-24-01536-f003:**
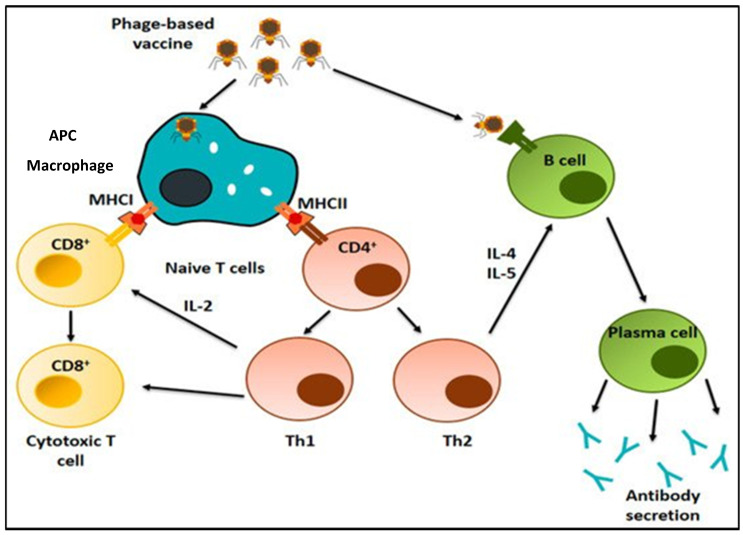
Phage-based vaccines stimulate both cellular and humoral immune responses. T cells are activated as a result of their presentation by MHC-I and MHC-II molecules. Phage particles may also activate the formation of plasma cells to secrete antibodies [36]. Reproduced with permission from [Jorge Benavides], [Vaccines]; published by [MDPI], [2020].

**Figure 4 ijms-24-01536-f004:**
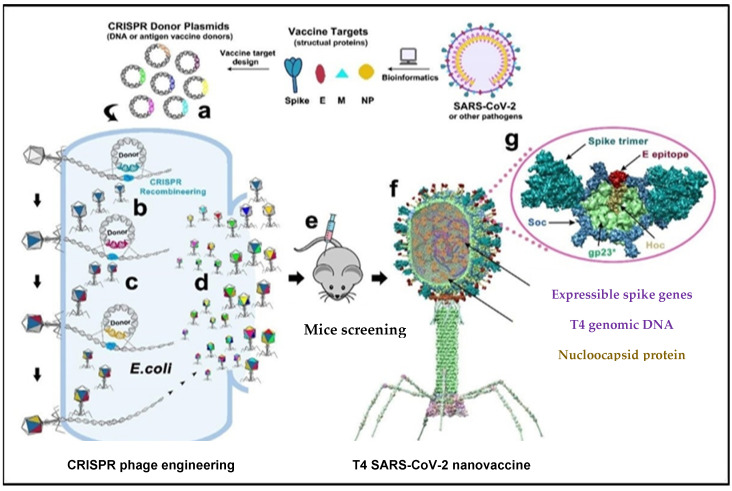
The genome of T4 phage was packed with engineered DNAs analogous to various components of SARS-CoV-2 virion. *E. coli* was grown and DNAs were inserted into it as a donor plasmid (a) CRISPR-targeted genome editing used for recpmbination into injected phage genome (b). Phage infections generated the several combinations of CoV-2 inserts and identified the recombinant phages in the progeny (c). For example, recombinant phage containing CoV-2 insert #1 (dark blue) can be used to infect CRISPR *E. coli* containing Co-V2 insert containing donor plasmid #2 (dark red). The progeny plaques obtained will contain recombinant phage #3 with both inserts #1 and #2 (dark blue plus dark red) in the same genome. This process was repeated to rapidly construct a pipeline of multiplex T4-SARS-CoV-2 vaccine phages (d). Selected vaccine candidates were then screened in a mouse model (e) to identify the most potent vaccine (f). Structural model of T4-SARS-CoV-2 Nanovaccine showing an enlarged view of a single 118 hexameric capsomer (g). The capsomer shows six subunits of major capsid protein gp23* (green), 119 trimers of Soc (blue), and a Hoc fiber (yellow) at the center of capsomer. The expressible spike genes are inserted into phage genome, the 12 aa E external peptide (red) is displayed at the tip of Hoc fiber, S-trimers (cyan) are attached to Soc subunits, and nucleocapsid proteins (yellow) are packaged in genome core. See Results and Materials and Methods for additional details. [28]. Reproduced with permission from [Venigalla B. Rao], [Science Advances]; published by [AAAS], [2021].

**Table 2 ijms-24-01536-t002:** Overview of phage-based vaccines/immunizations for COVID-19 that use different proteins of SARS-CoV-2 in different genera of bacteriophages.

Phage Genera	SARS-CoV-2 Proteins	Immune Responses	References
Mycobacteriophages	SARS-CoV-2 RB motif segments	Elicited IgG-mediated antibody responses in mice against SARS-CoV-2.	[23]
Universal bacteriophage T4	Envelope proteins and capsid proteins	The proteins were inserted into a phage using CRISPR engineering. Specific and broader immune responses were triggered against multiple SARS-CoV-2 targets using a single-phage backbone in which various antigens were incorporated.	[28]
12-mer phage	RB domain of the S protein	A phage display peptide library was screened against RBD, which showed binding ability with the RBD. It was demonstrated that peptide no. 1 could specifically bind to the SARS-CoV-2 RBD. It was confirmed that the SARS-CoV-2-specific peptide holds great promise as a new bioreceptor and ligand for the rapid and accurate detection of SARS-CoV-2.	[60]
Adeno-associated virus/phage (AAVP) particles	S protein	The vaccine showed a strong targeted humoral response. A strong systemic and specific immune response was observed with phage particles against SARS-CoV-2 in immune-competent mice by aerosol pulmonary vaccination.	[61]
Lambda phages	CoV-2-RBD of spike proteins	Robust immune responses were observed by the activity of hCoV-RBD decorated PLPs, which induced antibodies with neutralizing activities and presented the lambda system as a promising vaccine candidate with high versatility and robustness.	[62]
Phage (N0315S)	ORF1ab polyprotein, spike glycoprotein (S), ORF3a, envelope protein (E), membrane glycoprotein (M), ORF7a, ORF7b, ORF8 NP phosphoprotein (N), ORF10	A phage displayed peptide library identified B cell epitopes for SARS-CoV-2. In summary, it was concluded that this information regarding SARS-CoV-2 epitopes could help in the the rapid development of immunodiagnostic tools and showed promise for guiding epitope-based vaccine design.	[42]
Filamentous phage M13	RBD of SARS-CoV-2	A panel of phage mimotopes and spike receptor-binding protein AA clusters were identified which were able to bind to ACE2 and FGFR3. The phage probes that competed with viruses for cellular binding receptors could serve as leads in the development of vaccines.	[52]
T4 phage	Spike trimers on the exterior surface of the protein and the interior surface of the nucleocapsid protein	The vaccine showed valid results in providing complete protection against COVID-19. Different parts of the T4 phage genome had an enormous amount of unnecessary genetic space that could be ideally used for the development of a universal vaccine design template.	[63]

## Data Availability

Not applicable.

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
