# Peer review of "The Breadth of Bacteriophages Contributing to the Development of the Phage-Based Vaccines for COVID-19: An Ideal Platform to Design the Multiplex Vaccine"

_ijms, 2023, doi:10.3390/ijms24021536_

Round 1

Reviewer 1 Report

The prospect of utilizing phage as a vaccine scaffold is interesting. However, I still feel that this review only merely summarize recent works on phage-based vaccine and still do not provide any interesting insights on the development of technology. My suggestions and questions are as follow.

Main points:

A table summarizing the part of the SARS-CoV2 protein that got cloned and tested for phage-based vaccines could be useful for the reader. 

Which genera of phages have been tested? What is the advantage of phage-display vaccines over the displaying on  the VLP?

Is there any concern regarding in selecting the part that would be cloned and put on the phage particle?  Is there any size limitation? What kind of linker should be used and how long? What is the host suitable for production of the phage-based vaccine?

Is there concern regarding the folding as the S-protein is a membrane protein that require folding in ER and phage amplification is mainly occurred in prokaryotes?

Minor point
The figures should be made original. 

Author Response

The prospect of utilizing phage as a vaccine scaffold is interesting. However, I still feel that this review only merely summarize recent works on phage-based vaccine and still do not provide any interesting insights on the development of technology. My suggestions and questions are as follow.

Main points:

A table summarizing the part of the SARS-CoV2 protein that got cloned and tested for phage-based vaccines could be useful for the reader. 

  • Required table was inserted, thanks to the reviewer

Which genera of phages have been tested? What is the advantage of phage-display vaccines over the displaying on  the VLP?

  • We answered most of these concerns in the revised version of the manuscript

Is there any concern regarding in selecting the part that would be cloned and put on the phage particle?  Is there any size limitation? What kind of linker should be used and how long? What is the host suitable for production of the phage-based vaccine?

  • We answered most of these concerns in the revised version of the manuscript

Is there concern regarding the folding as the S-protein is a membrane protein that require folding in ER and phage amplification is mainly occurred in prokaryotes?

  • We answered most of these concerns in the revised version of the manuscript

Minor point

The figures should be made original. 

  • Due to issues related to license and so but we inserted references for everything

Reviewer 2 Report

Manuscript should be thouroughly checked by Authors, as there are minor errors - e.g. line 68 has an extra period in a sentence "...development during the COVID-19 pandemic. [18]."

The novelty of the paper is unclear - authors should expand the introduction to more closely explain the novelty presented by their manuscript.

Line 161.-162. - "Conclusively, phage display technology lets scientists take phages spaced out and put them back together like Lego blocks for vaccine development." - while I understand the metaphor, maybe a more serious one would be better.

Line 181., 212., 231., and many others - "et al." is written with a colon, as "al." is an abbreviation of "alia".

Some claims made by the authors would do well with an added citation, such as:

    Line 284.-287.: "Moreover, the engineering of the T4 bacteriophage by CRISPR (clustered regularly interspaced short palindromic repeats) paved the way for the rapid generation of the SARS-CoV-2 vaccine; however, previously it was considered to lack the flexibility of engineering."
    Line 304.-306.: "It is estimated that a few hundred thousand to 1,000,000 phage vaccines against SARS-CoV-2 doses can be formed in a 100-liter fermenter, thus leading to large-scale production."

Conclusions are very short - in this type of the paper they should be significantly wider, and offer a more detailed suggestions and repeat the key findings from the rest of the paper.

Author Response

Manuscript should be thouroughly checked by Authors, as there are minor errors - e.g. line 68 has an extra period in a sentence "...development during the COVID-19 pandemic. [18]."

  • We thank the reviewer for his notice, and we adjusted the referred sentence

The novelty of the paper is unclear - authors should expand the introduction to more closely explain the novelty presented by their manuscript.

  • We respect the reviewers concern, we mentioned in the abstract that engineering of the T4 bacteriophage by CRISPR presents phage vaccines as a valuable platform with potential capabilities of genetic plasticity, intrinsic immunogenicity, and stability

Line 161.-162. - "Conclusively, phage display technology lets scientists take phages spaced out and put them back together like Lego blocks for vaccine development." - while I understand the metaphor, maybe a more serious one would be better.

  • We paraphrased the sentence to be clearer and simpler

Line 181., 212., 231., and many others - "et al." is written with a colon, as "al." is an abbreviation of "alia".

  • References were revised

Some claims made by the authors would do well with an added citation, such as:

    Line 284.-287.: "Moreover, the engineering of the T4 bacteriophage by CRISPR (clustered regularly interspaced short palindromic repeats) paved the way for the rapid generation of the SARS-CoV-2 vaccine; however, previously it was considered to lack the flexibility of engineering."

    Line 304.-306.: "It is estimated that a few hundred thousand to 1,000,000 phage vaccines against SARS-CoV-2 doses can be formed in a 100-liter fermenter, thus leading to large-scale production."

  • We thank the reviewer for his suggestion, and we included suitable references

Conclusions are very short - in this type of the paper they should be significantly wider, and offer a more detailed suggestions and repeat the key findings from the rest of the paper.

  • We avoided repeating that much and we sum up key findings in the conclusion
